# Characteristics of patients with chronic idiopathic interstitial pneumonia undergoing repeated respiratory-related hospitalizations: A retrospective cohort study

**Ryo Yamazaki, Osamu Nishiyama** ⓘ *, **Sho Saeki, Hiroyuki Sano, Takashi Iwanaga, Yuji Tohda**

Department of Respiratory Medicine and Allergology, Kindai University, Faculty of Medicine, Osakasayama, Osaka, Japan

* nishiyama_o@yahoo.co.jp

**Data Availability Statement:** All relevant data are within the manuscript and its Supporting Information files.

## Abstract

### Background

Idiopathic pulmonary fibrosis (IPF) is a chronic and progressive lung disease. Chronic idiopathic interstitial pneumonia (c-IIP) is a group of lung diseases consisting mainly of fibrotic IIPs, and IPF is a type of c-IIP. Some patients with c-IIP undergo respiratory-related hospitalizations (RHs). With the hypothesis that the characteristics of patients who undergo RHs are related to the number of hospitalizations, we reviewed and investigated the RHs of patients with c-IIP.

### Methods

We retrospectively examined the data of patients with c-IIP who were admitted to Kindai University Hospital between January 2008 and December 2018 for respiratory-related causes.

### Results

During the study period, 243 patients with c-IIP were hospitalized a total of 544 times because of respiratory-related causes. The most common reasons for the first RH were acute exacerbation (48.1%) followed by pulmonary infection (32.5%). The most frequent reason for subsequent RHs was pulmonary infection. The in-hospital and 90-day mortality rate of patients with pulmonary infection increased with increasing numbers of RHs. Patients with multiple RHs had significantly worse long-term survival than patients hospitalized a single time.

### Conclusions

Pulmonary infection was the most frequent reason for repeated RHs. The proportion of all patients hospitalized for pulmonary infection at each RH increased with increasing numbers

**Funding:** The author(s) received no specific funding for this work.

**Competing interests:** The authors have declared that no competing interests exist.

of RHs, along with the mortality rate of patients with pulmonary infections. Furthermore, repeated RHs were associated with poor survival.

## Introduction

Idiopathic pulmonary fibrosis (IPF) is a chronic, progressive lung disease of unknown etiology[1]. The prognosis is poor, and the median survival is only 3 to 5 years from diagnosis; however, the natural history is variable [1, 2]. Many patients with IPF undergo acute respiratory events during their clinical course [3]. Respiratory-related hospitalization (RH) is an important factor in the outcomes of patients with IPF, because of its high in-hospital mortality, effect on subsequent survival, and the associated economic burden [3–5]. RHs of patients with fibrotic interstitial pneumonia are also significantly associated with in-hospital and post-discharge mortality [6] and are therefore important for predicting the long-term survival of patients with fibrotic interstitial pneumonia.

Several studies have reported that chronic respiratory insufficiency due to disease progression was the major cause of death in patients with IPF [7–9]. However, recent studies have demonstrated that pulmonary infection is a common reason for both RH and death [10, 11].

Some patients with IPF undergo several RHs during their clinical course. However, the characteristics of patients who undergo repeated RHs remain unknown. Identification of the characteristics of patients with IPF who require repeated RHs might lead to the development of appropriate prophylactic measures and treatments and a lengthened survival time.

Patients with a chronic form of idiopathic interstitial pneumonia (IIP) other than IPF are common. They have characteristics suggestive of IPF but are without a definitive diagnosis because surgical lung biopsy was not performed. They should be classified as patients with unclassifiable IIP [12]. However, in general practice, it may be relevant to consider patients with IPF together with patients with unclassifiable fibrotic IIP.

With the hypothesis that the characteristics of patients who undergo RHs are related to the number of hospitalizations, we retrospectively reviewed and investigated the RHs of patients with chronic IIP (c-IIP), including IPF.

## Methods

### Patients

We retrospectively examined the medical charts of all patients with c-IIP who were admitted to Kindai University Hospital for respiratory-related causes between January 2008 and December 2018. The definition of c-IIP was as follows: a chronic form of IIP that includes IPF. The diagnosis of IPF was based on recent guidelines [13]. Patients were excluded if they had other known causes of interstitial lung disease, as follows: domestic or occupational environmental exposure, connective tissue disease, and drug toxicity. Patients with a c-IIP other than IPF were those who were found to have a pattern of probable usual interstitial pneumonia (UIP) or indeterminate UIP on high-resolution computed tomography (HRCT), and were not subjected to surgical lung biopsy. Patients with an alternative diagnosis pattern on chest HRCT were excluded. HRCT patterns were categorized by pulmonologists with over 10 years of experience. For patients whose HRCT pattern was difficult to categorize, the final assessment was established after careful discussions between several specialists. Fig 1 shows the flow chart for inclusion and exclusion of patients who were hospitalized for the first time during the study period.

Patients with a chronic form of interstitial lung disease hospitalized for the first time during the study period    n= 1028

- Excluded for alternative diagnosis pattern on HRCT   n= 127

- Excluded for another known cause of interstitial lung disease such as environmental, domestic, or occupational exposure; connective tissue disease; or drug toxicity    n= 294

- Excluded for nonrespiratory-related and/or elective hospitalizations    n= 364

Included in the study    n=243

- Diagnosis of IPF    n=138

- Without diagnosis of IPF but with probable or indeterminate for UIP pattern on HRCT    n=105

**Fig 1. Flow chart of inclusion and exclusion criteria.** HRCT, high-resolution computed tomography; IPF, idiopathic pulmonary fibrosis; UIP, usual interstitial pneumonia.

Informed consent was waived, because this study was based on a retrospective analysis of case records from our university hospital. Approval for the use of these data for the analysis was provided by the ethics committee of the Kindai University Faculty of Medicine (No. 31–044).

### Diagnosis of acute exacerbation and pulmonary infection

Acute exacerbation of c-IIP was defined in reference to a recent international working group report for acute exacerbation of IPF [14] as follows: 1) a previous or concurrent diagnosis of c-IIP; 2) acute worsening or development of dyspnea typically of less than 1 month's duration; 3) HRCT finding of new bilateral ground-glass opacities and/or consolidation superimposed on a background consistent with the UIP, probable UIP, or indeterminate for UIP pattern; and 4) deterioration not fully explained by cardiac failure or fluid overload. The diagnosis of acute exacerbation of c-IIP was made in reference to these criteria. Pneumonia was defined as

follows: 1) newly developed consolidation and/or ground-glass opacities on chest X-ray or HRCT; and 2) clinical evidence such as fever, productive cough, chills, or abnormal white blood cell count. If the findings on chest radiography and/or HRCT had not changed from previous findings, the diagnosis was considered to be bronchitis. For patients in whom a diagnosis was difficult to establish, the final diagnosis was made after careful discussion involving several specialists (pulmonologists, each with over 10 years of experience).

## Pulmonary function tests

Pulmonary function tests that had been performed within 1 year prior to admission were used for determining baseline pulmonary function. Pulmonary function tests, including spirometry and single-breath measurements of the diffusing capacity of the lungs for carbon monoxide (DLco) (CHESTAC-8800; Chest, Tokyo, Japan), were performed according to the standards of the European Respiratory Society [15, 16]. Results were expressed in absolute values and as percentages of normal predictive Japanese values [17, 18].

## Data collection

We evaluated the following clinical characteristics of every patient: age, smoking status, long-term oxygen therapy (LTOT), and treatment for c-IIP. Standard laboratory testing and routine blood sampling had been performed on admission.

## Respiratory-related hospitalization

RH was defined as a nonelective hospitalization due to an acute respiratory event that included acute exacerbation of c-IIP, pulmonary infection (pneumonia or bronchitis), pneumothorax and/or mediastinal emphysema, heart failure, and pulmonary embolism. All RHs during the study period were included in the study; however, nonrespiratory-related and any elective hospitalizations were excluded. All hospitalizations were retrospectively confirmed during reviews of the hospital charts.

## Assessment of survival

We assessed in-hospital and 90-day mortalities. In-hospital mortality was defined as the percentage of patients who died during the corresponding hospitalization without discharge (mean duration of hospitalization: 33.7±41.1, 26.4±19.4, 22.0±13.1, and 24.3±20.9 days at first, second, third, and ≥fourth hospitalizations, respectively). Total survival was also assessed through April 24, 2019. All deaths were retrospectively confirmed during reviews of the hospital charts. If a patient was still alive, he or she was treated as a censored case in the analysis. The total survival time from the first day of each respiratory-related hospitalization was also determined.

## Statistical analysis

Continuous variables are presented as means±standard deviation (SD). Categorical variables are expressed as frequencies. Comparisons of categorical variables were performed by the Fisher exact test. Comparisons of parameters for the causes of hospitalization were made by one-way analysis of variance, followed by the Bonferroni correction for multiple comparisons. Kaplan-Meier survival curves were constructed based on the frequency of RHs, and compared by the log-rank test. All tests were performed with a significance level of less than 0.05. Analyses were performed by the PASW statistical package, ver.18 (SPSS Japan Inc., Tokyo, Japan).

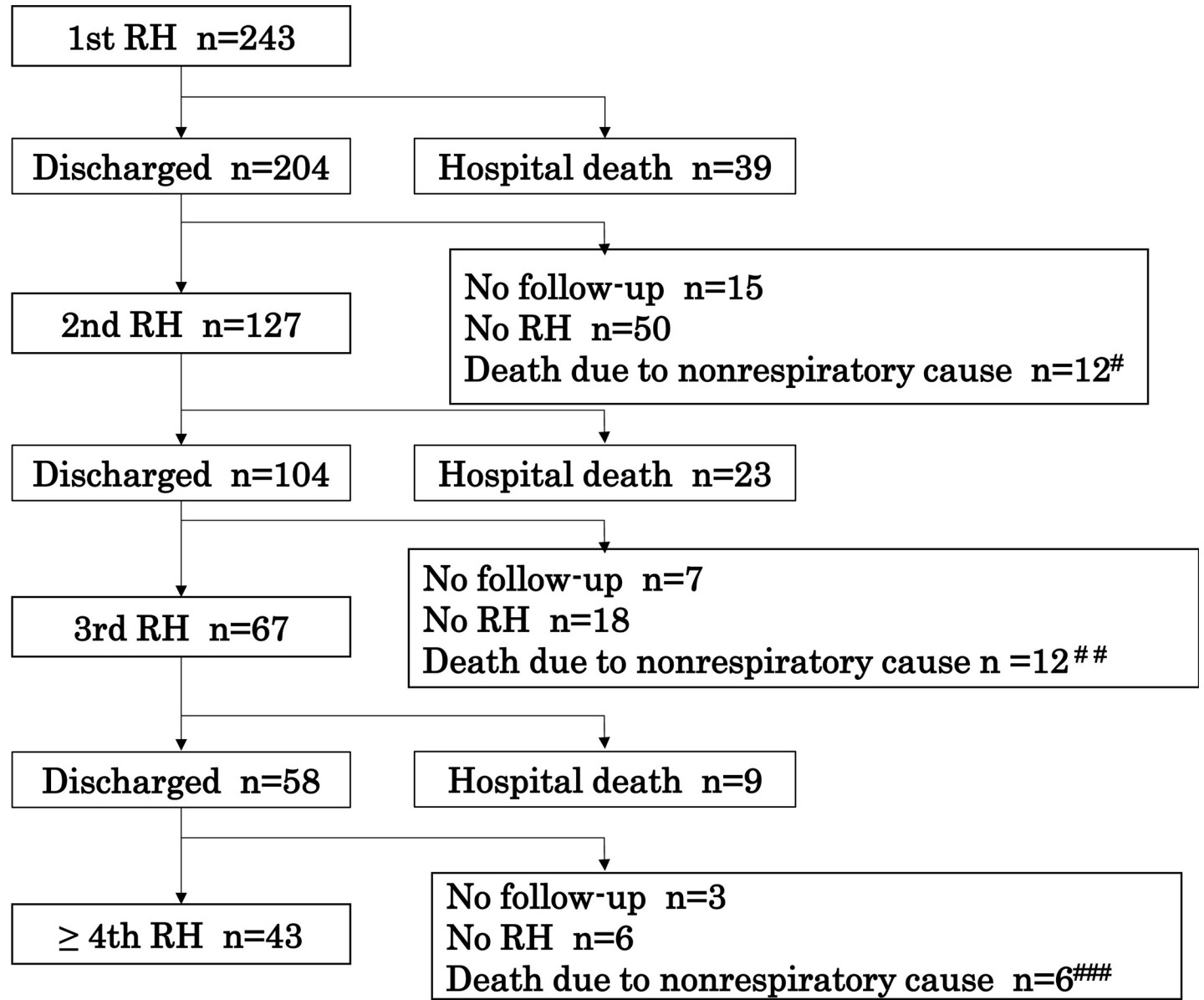

**Fig 2. Hospitalization flow chart.** #: 9 unknowns, 2 lung cancers, and 1 esophageal cancer, ##: 8 unknowns, 1 lung cancer, 1 malignant lymphoma, 1 peritonitis, and 1 cerebral infarction, ###: 3 unknowns, 1 hepatic cell carcinoma, 1 ileus, and 1 suicide. RH, respiratory-related hospitalization.

## Results

During the study period, 243 patients (177 men and 66 women) with c-IIP were hospitalized a total of 544 times for respiratory-related causes at the University Hospital. As shown in Fig 2, of 243 patients who were hospitalized for the first time, 39 (16.0%) died in the hospital. Of 204 patients (84.0%) who were discharged, 15 (6.2%) were not followed-up, 50 (20.6%) were followed-up without any RH, 12 (4.9%) died of nonrespiratory-related causes, and 127 (52.3) underwent a second RH during the period of observation. Fig 2 also shows the distribution of patients after the second RH.

The baseline clinical characteristics of the patients before the first hospitalization are shown in Table 1. The mean age was 74.8±7.3 years. The mean forced vital capacity (FVC) was 73.5

**Table 1. Baseline characteristics of patients before the first hospitalization.**

| Characteristics | All patients |
|---|---|
| | **N = 243** |
| Age, yr | 74.8 ± 7.3 |
| Gender | |
| Male/Female | 177/66 |
| Body mass index | 22.2 ± 4.0 [a] |
| Pulmonary function tests | |
| FVC, L | 2.1 ± 0.7 [b] |
| FVC, % predicted | 73.5 ± 22.4 [b] |
| DLco, mL/min/mmHg | 9.3 ± 3.3 [c] |
| DLco, % predicted | 65.0 ± 20.8 [c] |
| Smoking status | |
| Current/Former/Never | 10/160/73 |
| Definitive IPF/others | 138/105 |
| Treatment for c-IIP at baseline | |
| Pirfenidone | 17 |
| Nintedanib | 9 |
| Corticosteroid | 13 |
| Cyclosporine | 5 |
| Cyclophosphamide | 2 |
| Tacrolimus | 1 |
| None | 201 |
| Long-term oxygen therapy | |
| yes/no | 47/196 |

Values are shown with actual number or as means±standard deviation.

a: n = 212

b: n = 148

c: n = 97

DLco = diffusing capacity for carbon monoxide, c-IIP = chronic idiopathic interstitial pneumonia, FVC = forced vital capacity, IPF = idiopathic pulmonary fibrosis.

±22.4% predicted, and mean DLco was 65.0±20.8% predicted. Of 243 patients with c-IIP, 138 had received a definitive diagnosis of IPF; IPF was diagnosed from histopathological findings on a biopsy specimen in 15 patients and without findings from a surgical biopsy in 123 patients. The other 105 patients had received a diagnosis of c-IIP, but not a definitive diagnosis. Because none of these 105 patients underwent a surgical lung biopsy, they were diagnosed with unclassifiable IIP. There were no patients who were pathologically diagnosed with idiopathic non-specific interstitial pneumonia, desquamative interstitial pneumonia, or respiratory bronchiolitis-interstitial lung disease. An antifibrotic agent was being used to treat 26 of 243 (10.7%) patients.

The baseline clinical characteristics before the first hospitalization of patients stratified by 5 different causes of RH are shown in Table 2. The differences between the values for body mass index (BMI) and FVC, percent predicted, were significant ($P < 0.0001$ and $P = 0.01$, respectively). By intergroup comparisons, patients hospitalized for pneumothorax and/or mediastinal emphysema had significantly lower values for BMI and FVC percent predicted than patients hospitalized with acute exacerbation or pulmonary infection (BMI: $P < 0.0001$ and $P < 0.001$, respectively; and FVC percent predicted: $P = 0.002$ and $P = 0.001$, respectively). The

**Table 2. Baseline characteristics of patients before the first hospitalization by cause.**

| Characteristics | Acute exacerbation | Pulmonary infection | Pneumothorax and/or mediastinal emphysema | Heart failure | Others[‖] | |
|---|---|---|---|---|---|---|
| **Patients** | **117** | **79** | **25** | **8** | **14** | **P value** |
| Age, yr | 74.3 ± 8.3 | 75.0 ± 6.6 | 74.5 ± 5.4 | 78.6 ± 5.7 | 77.1 ± 5.6 | 0.39 |
| Gender; Male/Female | 82/35 | 58/21 | 20/5 | 6/2 | 11/3 | 0.85 |
| Body mass index | 23.4 ± 3.7 [a] | 21.9 ± 3.7 [b] | 18.9 ± 4.0 [c] * | 20.9 ± 5.1 | 22.3 ± 4.0 | <0.0001 |
| Pulmonary function test | | | | | | |
| FVC, L | 2.2 ± 0.7 [d] | 2.2 ± 0.8 [e] | 1.7 ± 0.5 [f] | 2.4 ± 0.6[g] | 1.9 ± 0.8 [h] | 0.16 |
| FVC, %predicted | 75.0 ± 18.5 [d] | 77.3 ± 24.9 [e] | 56.2 ± 18.1 [f †] | 75.7 ± 14.5 [g] | 67.4 ± 38.4 [h] | 0.01 |
| DLco, mL/min/mmHg | 10.1 ± 3.9 [i] | 8.9 ± 2.6 [j] | 7.6 ± 0.7 [k] | 6.9 ± 1.6 [g] | 11.1 ± 0.1 [l] | 0.05 |
| DLco, % predicted | 69.0 ± 23.4 [i] | 60.9 ± 17.0 [j] | 62.1 ± 11.6 [k] | 56.1 ± 29.2[g] | 84.7 ± 5.0 [l] | 0.19 |
| Smoking status; yes/no | 78/39 | 60/19 | 18/7 | 6/2 | 8/6 | 0.52 |
| LTOT; yes/no | 12/105 | 25/54 [#] | 4/21 | 2/6 | 4/10 | 0.004 |

Values are shown as actual number or means±standard deviation.

‖ Others included 6 hemoptysis, 5 chronic respiratory failure, 2 drug-induced lung injury, and 1 pulmonary embolism

* $P < 0.0001$ and $P < 0.001$ compared with acute exacerbation group and pulmonary infection group, respectively.

† $P = 0.002$ and $P = 0.001$ compared with acute exacerbation group and pulmonary infection group, respectively.

# $P = 0.0003$ compared with acute exacerbation group.

a: n = 98

b: n = 73

c: n = 22

d: n = 71

e: n = 50

f: n = 16

g: n = 6

h: n = 6

i: n = 45

j: n = 37

k: n = 8

l: n = 2

DLco = diffusing capacity for carbon monoxide, FVC = forced vital capacity, IPF = idiopathic pulmonary fibrosis, LTOT = Long-term oxygen therapy.

differences between the proportions of patients receiving LTOT grouped according to cause of RH were significant ($P = 0.004$). By intergroup comparison, the proportion of admitted patients with pulmonary infection who received LTOT was higher than the proportion of admitted patients with acute exacerbation who received LTOT ($P = 0.0003$).

Table 3 shows the causes of RH and the patient mortality rates grouped according to which hospitalization (first, second, third, or fourth or more). The most frequent cause of the first hospitalization was acute exacerbation (48.1%) followed by pulmonary infection (32.5%). The most frequent cause for all subsequent hospitalizations was pulmonary infection. As the number of RHs increased, the proportion of admissions for pulmonary infection increased and the proportion of admissions for acute exacerbation decreased. The proportions of admissions for pneumothorax and/or mediastinal emphysema and heart failure did not change based on whether or not a patient was hospitalized an increasing number of times. Among the 243 first hospitalizations, the in-hospital and 90-day mortality rates for all causes were 14.8% and 19.3%, respectively. The in-hospital and 90-day mortality rates of patients hospitalized for pulmonary infection increased with increasing number of RHs, whereas the rates did not increase among patients hospitalized increasing numbers of times for acute exacerbation.

**Table 3. Causes of the hospitalization and mortality of each hospitalization.**

| Causes | N (%) | Mortality (%) | |
|---|---|---|---|
| | | In-hospital | 90-day |
| First hospitalization | | | |
| Total | 243 | 14.8% | 19.3% |
| Acute exacerbation | 117 (48.1%) | 22.2% | 23.9% |
| Pulmonary infection | 79 (32.5%) | 6.3% | 13.9% |
| Pneumothorax and/or mediastinal emphysema | 25 (10.2%) | 16.0% | 20.0% |
| Heart failure | 8 (3.2%) | 0% | 12.5% |
| Others ‖ | 14 (5.7%) | 7.1% | 14.2% |
| Second hospitalization | | | |
| Total | 127 | 17.3% | 21.2% |
| Acute exacerbation | 31 (24.4%) | 29.0% | 25.8% |
| Pulmonary infection | 57 (44.8%) | 12.2% | 15.7% |
| Pneumothorax and/or mediastinal emphysema | 17 (13.3%) | 29.4% | 29.4% |
| Heart failure | 11 (8.6%) | 8.3% | 33.3% |
| Others * | 11 (8.6%) | 0% | 9.0% |
| Third hospitalization | | | |
| Total | 67 | 13.4% | 17.9% |
| Acute exacerbation | 9 (13.4%) | 33.3% | 33.3% |
| Pulmonary infection | 38 (56.7%) | 10.5% | 18.4% |
| Pneumothorax and/or, mediastinal emphysema | 7 (10.4%) | 14.2% | 14.2% |
| Heart failure | 11 (16.4%) | 0% | 0% |
| Others † | 2 (2.9%) | 50.0% | 50.0% |
| ≥ fourth hospitalization | | | |
| Total | 107 | 12.1% | 21.4% |
| Acute exacerbation | 9 (8.4%) | 33.3% | 33.3% |
| Pulmonary infection | 75 (70.0%) | 9.3% | 21.3% |
| Pneumothorax and/or, mediastinal emphysema | 14 (13.0%) | 14.2% | 14.2% |
| Heart failure | 7 (6.5%) | 0% | 14.2% |
| Others # | 2 (1.8%) | 50.0% | 50.0% |

Values are shown with actual number or percentage.

Some patients overlap. For example, a patient who underwent 2 respiratory-related hospitalizations during the study period was included in the analyses of both the first and second hospitalization.

‖ 6 hemoptysis, 5 chronic respiratory failure, 2 drug-induced lung injury, and 1 pulmonary embolism

* 5 chronic respiratory failure, 3 hemoptysis, 2 pulmonary embolism, and 1 drug-induced lung injury

† 1 pulmonary embolism and 1 chronic pulmonary aspergillosis

# 1 pulmonary embolism and 1 lymphangitic carcinomatosis

The Kaplan-Meier survival curves of patients grouped according to the ordinal number of hospitalization are shown in Fig 3. In this analysis, the last hospitalization we recognized during the study period was used for categorization, so that none of the patients overlapped. Survival after the first hospitalization was significantly better than the survival of patients after multiple hospitalizations (log rank, $P = 0.004$). However, the differences between the survival of patients at the second RH and each subsequent RH were not significant. The estimated median duration of survival was 1285, 88, 104, and 22 days after the first, second, third, and fourth or greater RH, respectively.

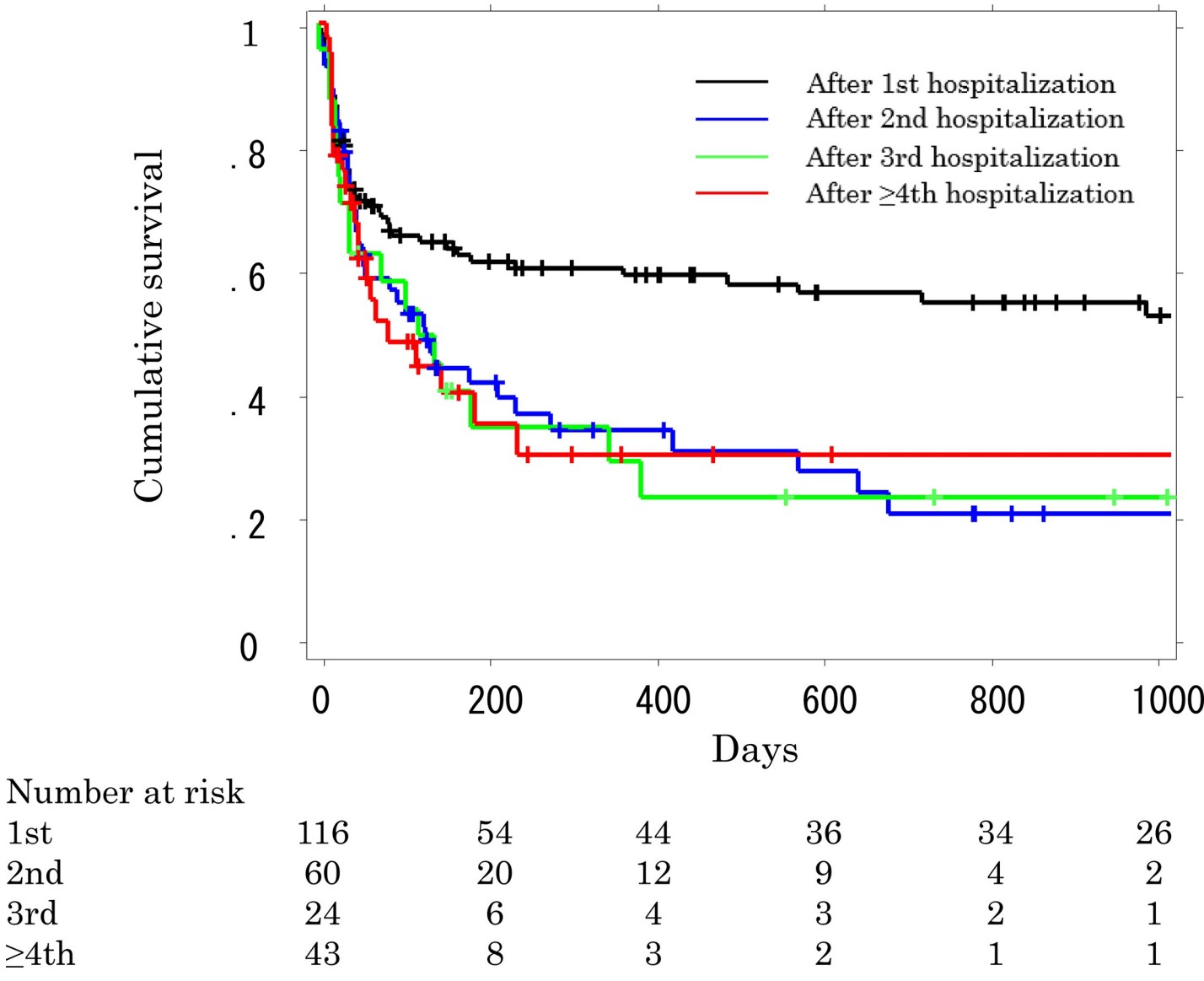

**Fig 3. Kaplan-Meier survival curves of each respiratory-related hospitalization identified by ordinal number.** Patients who underwent 1, 2, 3, and 4 or more hospitalizations during the study period were categorized into "After the first, second, third, and ≥ fourth hospitalization" group, so that no patient overlapped. Survival curves were compared by the log-rank test ($P$ = 0.004). +: censored. RH, respiratory-related hospitalization.

## Discussion

This study of patients with c-IIP who underwent RHs evaluated and reported their characteristics not only at the first admission but at each repeated hospitalization. The mortality of patients who were hospitalized because of respiratory-related causes was high. Pulmonary infections, as well as acute exacerbations, accounted for a large proportion of the causes of RH in patients with c-IIP. The mortality rates of patients with pulmonary infection increased as the number of RHs increased, whereas that was not the case with patients admitted for acute exacerbations. Furthermore, the estimated long-term mortality of patients was worse after the second RH and subsequent RHs compared to the long-term mortality of patients after the first hospitalization, although no differences were observed for in-hospital and 90-day mortality rates.

An acute exacerbation of c-IIP is an important clinical event in the course of patients with c-IIP. For patients with IPF, it is known that their prognosis after acute exacerbation is extremely poor [3]. Actually, the most frequently reported cause of death in patients with IPF is acute exacerbation [7], with the other reported cause of death being chronic respiratory failure due to disease progression [7–9]. However, in a recent study evaluating discharge summaries at a French hospital, the most frequent reason for the RH of patients with IPF was pulmonary infection, which accounted for 43.7% of all hospitalized patients, followed by acute respiratory worsening, which accounted for 36.5% [5]. This result is consistent with our study. Moreover, a previous study by our group also demonstrated the importance of pulmonary infection as a reason for a RH in patients with IPF [11]. Therefore, pulmonary infection might have been underestimated as a reason for RHs and death in patients with c-IIP, including those with IPF.

Another important result of our study is that the proportion of all patients hospitalized for pulmonary infection at each RH increased with increasing number of RHs. It can be said that pulmonary infection is increasingly important in patients undergoing repeated RHs. To our knowledge, this study is the first to have clarified the reasons for RH and mortality in c-IIP patients based on the number of hospitalizations.

Although RH itself leads to high mortality in patients with IPF, subsequent survival is also poor, even if a patient has survived the RH [4, 19]. Presumably RH adversely affects a patient's lung function, which results in decreased activities of daily living (ADL). Decreased lung function and decreased ADL are associated with poor survival [20, 21]. Repeated RHs seem to more adversely affect subsequent long-term survival than short-term survival. This conclusion is apparent from the survival curves, which show that the survival curves of the second RH and subsequent RHs indicate similarly worse survival than the curve after the first RH, although lead-time bias should be taken into account. However, the in-hospital and 90-day mortalities were comparable after every RH. Preventing an acute respiratory event associated with c-IIP may be another challenge for the prolongation of survival, especially for patients who have already been hospitalized a single time for a respiratory cause.

This study found that acute exacerbation was the most frequent cause for the first RH of patients with c-IIP. However, pulmonary infection became the most frequent cause after the second RH. One possible reason for this phenomenon is that patients who are hospitalized with acute exacerbation are treated with steroids and/or immunosuppressing agents, which can lead to subsequent RHs for pulmonary infection. The relationship between tapering immunosuppressive treatment and subsequent risk for pulmonary infection should be investigated in further studies.

The strength of our study includes the relatively large number of hospitalized patients with c-IIP from a real-world clinical practice. The 10-year study period was sufficient for enrolling the patients and for estimating long-term survival rates.

This study has limitations. First, it was a single-center study. Since standards for hospitalization can differ between regions and countries, a multicenter study would be preferable. Second, this was a retrospective study. Third, patients with interstitial lung disease that was not idiopathic might have been included in the study, although an effort was made to exclude patients with collagen vascular disease, chronic hypersensitivity pneumonia, drug-related interstitial lung disease, and so on. Moreover, given that the present study cohort might consist of patients with a variety of fibrotic interstitial lung diseases, and that the treatment regimen was not consistent among the patients or for each type of disease, caution is needed when generalizing the results. We defined a patient with c-IIP as a patient with a diagnosis of IPF or with an HRCT pattern of probable or indeterminate for UIP. The term "c-IIP" has not been validated, although it has been used in several studies [22,23]. Actually, the patients with c-IIPs

other than IPF who were included in the study corresponded to those with unclassifiable IIP, which was defined by an official 2013 American Thoracic Society/European Respiratory Society statement [12]. The concepts of c-IIP and unclassifiable IIP need further refinement. Finally, distinguishing between acute exacerbation and pulmonary infection was difficult for some cases, even with considerable effort exerted for the differential diagnosis. Acute exacerbation and pulmonary infection might even overlap in some cases.

In conclusion, repeated RHs are usually observed in patients with c-IIP. Not only acute exacerbation but also pulmonary infection account for the major reasons for repeated RH. The proportion of all patients hospitalized for pulmonary infection at each RH increased with increasing number of RHs, and the proportion of patients hospitalized for acute exacerbation decreased. The mortality rate of patients with pulmonary infection increased with increasing numbers of RHs, although the mortality rate of patients with acute exacerbations did not increase. Repeated RH is associated with subsequent poor long-term mortality.

## Supporting information

**S1 Data.**
(XLSX)

## Author Contributions

**Conceptualization:** Ryo Yamazaki, Osamu Nishiyama.

**Data curation:** Ryo Yamazaki, Sho Saeki.

**Formal analysis:** Ryo Yamazaki, Osamu Nishiyama.

**Investigation:** Ryo Yamazaki.

**Methodology:** Osamu Nishiyama.

**Project administration:** Yuji Tohda.

**Supervision:** Osamu Nishiyama.

**Writing – original draft:** Ryo Yamazaki.

**Writing – review & editing:** Osamu Nishiyama, Hiroyuki Sano, Takashi Iwanaga.

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
