## [Decision Letter · Decision Letter 0]

3 Dec 2019

PONE-D-19-28502

Characteristics of repeated respiratory-related hospitalizations in patients with fibrosing idiopathic interstitial pneumonia: A retrospective cohort study

PLOS ONE

Dear Dr. Nishiyama,

Thank you for submitting your manuscript to PLOS ONE. After careful consideration, we feel that it has merit but does not fully meet PLOS ONE’s publication criteria as it currently stands. Therefore, we invite you to submit a revised version of the manuscript that addresses the points raised during the review process.

The input of two content-specific expert reviewers was received on your original submission, with their comments outlined below. There are some significant concerns raised through the peer review process which in this instance you are being given an opportunity to consider should you wish to pursue re-submission. 

We would appreciate receiving your revised manuscript by Jan 17 2020 11:59PM. To enhance the reproducibility of your results, we recommend that if applicable you deposit your laboratory protocols in protocols.io, where a protocol can be assigned its own identifier (DOI) such that it can be cited independently in the future. For instructions see: http://journals.plos.org/plosone/s/submission-guidelines#loc-laboratory-protocols

We look forward to receiving your revised manuscript.

Kind regards,

Shane Patman, PhD

Academic Editor

PLOS ONE

Journal Requirements:

2. In the ethics statement in the manuscript and in the online submission form, please provide additional information about the patient records used in your retrospective study, including: a) whether all data were fully anonymized before you accessed them and b) the date range (month and year) during which patients' medical records were accessed.

Additional Editor Comments (if provided):

Reviewers' comments:

Reviewer's Responses to Questions

**Comments to the Author**

1. Is the manuscript technically sound, and do the data support the conclusions?

Reviewer #1: Partly

Reviewer #2: Partly

2. Has the statistical analysis been performed appropriately and rigorously? 

Reviewer #1: I Don't Know

Reviewer #2: No

3. Have the authors made all data underlying the findings in their manuscript fully available?

Reviewer #1: Yes

Reviewer #2: Yes

4. Is the manuscript presented in an intelligible fashion and written in standard English?

Reviewer #1: Yes

Reviewer #2: No

5. Review Comments to the Author

Reviewer #1: Nishimura et al. describe the prognosis of patients with chronic fibrosing idiopathic interstitial pneumonia (f-IIP) and the reasons of their hospital admissions. The data of long-term follow-up for ten years in patients with fibrosing interstitial pneumonias is interesting, however, because of the obscure definitions and criteria of interstitial pneumonia, it is somewhat hard to understand the results and discussions of this study. Therefore, several concerns are necessary to be revised as described elsewhere.

Major comments:

1. The term “f-IIP” may not be commonly used, and the relationship between usually used “IIPs” and “fIIP” is unclear. Please add more information regarding the criteria used to make a diagnosis and categorization of chronic f-IIP. The definitions and criteria of ATS for interstitial pneumonia other than IPF may also be suitable for describing this categorization of patients with interstitial pneumonia. A table with the information of patients in each category and the results of RH is informative.

2. Chronic hypersensitivity pneumonia is always a difficult differential diagnosis of IPF. How did the authors exclude CHP? IN addition, was MDD applied for the diagnosis?

3. In the methods, it was hard to understand the entry and exclusion criteria of this study. The authors should thus make a flow chart while clearly showing the entry and exclusion criteria.

4. Kaplan-Meier survival curves were used based on the frequency of RHs. However, according to Figure 2, it seemed as though each group had duplicate patients, and the total number of patients in each group should be 243 patients for Kaplan-Meier method.

5. IPF is not a focus of this study, and please concentrate on the focused disease entity, non-IPF interstitial pneumonias, in introduction and other sections, not on IPF.

Minor comments:

Page 7, Line 13

“For patients in whom a diagnosis was difficult to establish, the final diagnosis was made after careful discussion involving several specialists.”

-> Please add more precise information about “specialists” in a certain field.

Page 8, Line 15

The study period and entry and exclusion criteria should be described.

Reviewer #2: This is a retrospective study to investigate the association among patient characteristics, respiratory-related hospitalizations (RHs), and prognosis in patients with chronic fibrosing interstitial pneumonia (f-IIP). The authors concluded that pulmonary infection was the most common respiratory-related event associated with hospitalization in patients with f-IIP and the mortality rate in those patients with pulmonary infection increased with increased number of RHs. The study concept is interesting. However, there are several critical drawbacks with respect to study design, inclusion and exclusion criteria, heterogeneity of IIPs included in this study, and statistical method.

Abstract

- “Chronic fibrosing idiopathic interstitial pneumonia (f-IIP) is a chronic and progressive lung disease.” This sentence is misleading because f-IIP is not a distinct disease but a group of interstitial lung diseases. The authors should amend this appropriately.

Introduction

- Page 4, line 16-17; “probably because physicians who do routine check-ups for IPF patients do not usually treat RHs in Western countries”

This is just the authors’ conjecture. The authors should cite some appropriate papers in this manuscript.

- Page 5, line 5-6; “Identification of the characteristics of IPF patients who require repeated RHs might be useful for prolonging their survival.”

How can the identification of the characteristics contribute to improvement of patients’ survival? The reviewer think that there is a logical gap between the identification and prolonging patients’ survival. More explanation on this should be given in the introduction part.

Methods

Patients

- The authors did not present inclusion criteria or exclusion criteria in the methods part. Furthermore, in the main body and Tables, there was no description with respect to ILD diagnoses included in this study.

- This study cohort consisted of a variety of ILDs and the treatment regimen was not consistent among the patients or diseases. Therefore, it is difficult to generalize the results of the study.

Diagnosis of acute exacerbation and pulmonary infection

- In this study, the definition of acute exacerbation (AE) of IPF was based on the 2016 international working group report, which was broader than former criteria. This 2016 definition accepts the concept of triggered-AE including infection. Therefore, in the present study, AE may have overlapped with pulmonary infection. I wonder how the authors distinguished between AE and pulmonary infection exhibiting bilateral consolidation and/or ground-glass opacities on HRCT.

Respiratory-related hospitalization

- Is “heart failure” an acute respiratory event? Please explain why it was included.

Assessment of survival

- Please define in-hospital mortality in this part and show the median (range or IQR) or mean (SD) days.

Results

- Figure 2 was based on the survival time from the date of the first, second, third, or ≥ fourth RH until the date of death or last visit, which is associated with a kind of lead-time bias. Furthermore, patients with multiple hospitalization probably belong to not only the 1st RH group but also ≥2nd RH groups. The 1st, 2nd, 3rd, and ≥4th RH groups were not independent. Therefore, these statistical methods are incorrect.

Discussion

- Page 21, line 9-11; “Preventing hospitalization of patients with f-IIP is crucial for prolonging their survival, especially for patients who have already been hospitalized once for a respiratory cause”.

This sentence sounds strange. Presumably, preventing an acute respiratory event is crucial.

6. PLOS authors have the option to publish the peer review history of their article (what does this mean?). If published, this will include your full peer review and any attached files.

Reviewer #1: No

Reviewer #2: No

---

## [Author Response · Author response to Decision Letter 0]

15 Jan 2020

Dear Dr. Patman:

Thank you for your kind and careful review of our manuscript that we submitted for publication. We have tried to address all of the reviewers' concerns. In addition, we made some grammatical change according to a native speaker’s check including the title.

To Reviewer #1

For the major comments

1. The term “f-IIP” may not be commonly used, and the relationship between usually used “IIPs” and “fIIP” is unclear. Please add more information regarding the criteria used to make a diagnosis and categorization of chronic f-IIP. The definitions and criteria of ATS for interstitial pneumonia other than IPF may also be suitable for describing this categorization of patients with interstitial pneumonia. A table with the information of patients in each category and the results of RH is informative.

Response: We thank the reviewer for this valuable comment. According to the reviewer’s suggestion, we changed the explanation on inclusion and exclusion criteria as follows (P6,L5): “The definition of f-IIP was as follows: a chronic form of idiopathic interstitial pneumonia (IIP) that includes IPF and IIP suggestive of IPF but without a definitive diagnosis. The diagnosis of IPF was based on recent guidelines [12]. Patients were excluded if they had other known causes of interstitial lung disease, as follows: domestic or occupational environmental exposure, connective tissue disease, and drug toxicity. Patients with IIP suggestive of IPF but without a definitive diagnosis were patients who were found to have a pattern of probable usual interstitial pneumonia (UIP) or indeterminate UIP on high-resolution computed tomography (HRCT), and were not subjected to surgical lung biopsy. Fig 1 shows the flow chart for exclusion and inclusion.”

We also added following information to Table 1 to clarify the breakdown: “Definitive IPF/others 138/105”. We also added the following (P10,L6): “Of 243 patients with f-IIP, 138 had received a definitive diagnosis of IPF; IPF was diagnosed from histopathological findings on a biopsy specimen in 15 patients and without findings from a surgical biopsy in 123 patients.” 

2. Chronic hypersensitivity pneumonia is always a difficult differential diagnosis of IPF. How did the authors exclude CHP? In addition, was MDD applied for the diagnosis?

Response: We diagnosed CHP when corresponding histopathological findings were observed on SLB. We also diagnosed CHP in patients without SLB by chest HRCT findings, including abnormality with predominance in the upper lungs and predominant findings of air trapping on HRCT, and BALF findings of markedly elevated lymphocyte count. We performed MDD when SLB was obtained, but the diagnosis was finalized after discussion with selected pulmonologists when SLB was not obtained. 

3. In the methods, it was hard to understand the entry and exclusion criteria of this study. The authors should thus make a flow chart while clearly showing the entry and exclusion criteria.

Response: According to the reviewer’s suggestion, we created a flow chart showing the exclusion and inclusion criteria (Fig. 1).

4. Kaplan-Meier survival curves were used based on the frequency of RHs. However, according to Figure 2, it seemed as though each group had duplicate patients, and the total number of patients in each group should be 243 patients for Kaplan-Meier method.

Response: Patients who underwent 1, 2, 3, and 4 or more hospitalizations during the study period were categorized into “After the first, second, third, or ≥ fourth hospitalization” groups. The last hospitalization we recognized in the study period was used for categorization in the Kaplan-Meier curves, so that none of the patients overlapped. We added the following sentence to explain this (P18,L2): “In this analysis, the last hospitalization we recognized in the study period was used for the categorization, so that no patients overlapped.”.

To avoid misunderstanding, we also added the following at the bottom of the Table 3: “Some patients overlap. For example, a patient who underwent 2 respiratory-related hospitalizations during the study period was included in the analyses of both the first and second hospitalization.”

We also added the number of patients at risk to Fig. 3. 

5. IPF is not a focus of this study, and please concentrate on the focused disease entity, non-IPF interstitial pneumonias, in introduction and other sections, not on IPF.

Response: Thank you for your valuable comments. But actually, we wanted to focus mainly on IPF. We also think that the patients without a definitive diagnosis of IPF who had a probable or indeterminate UIP pattern on chest HRCT had a high possibility of having IPF in this study. So, we mainly discussed IPF in the Introduction and Discussion sections.

Minor comments:

Page 7, Line 13

“For patients in whom a diagnosis was difficult to establish, the final diagnosis was made after careful discussion involving several specialists.”

-> Please add more precise information about “specialists” in a certain field.

Response: We added the following sentence to explain this (P7,L12): “(pulmonologists, each with over 10 years of experience)”.

Page 8, Line 15

The study period and entry and exclusion criteria should be described.

Response: The study period was between January 2008 and December 2018. That information is in the Methods section (P6,L4). We added the following sentence to explain the inclusion and exclusion criteria (P8,L12): “All RHs during the study period were included in the study; however, nonrespiratory-related and any elective hospitalizations were excluded.” 

We also added the following to note the period for total survival time (P9,L1): “Total survival was assessed through April 24, 2019.”.

To Reviewer #2.

Abstract

- “Chronic fibrosing idiopathic interstitial pneumonia (f-IIP) is a chronic and progressive lung disease.” This sentence is misleading because f-IIP is not a distinct disease but a group of interstitial lung diseases. The authors should amend this appropriately.

Response: According to the reviewer’s suggestion, we changed the sentence as follows (P2,L3): “Idiopathic pulmonary fibrosis (IPF) is a chronic and progressive lung disease. Chronic fibrosing idiopathic interstitial pneumonia (f-IIP) is a group of fibrotic IIPs, and IPF is a type of chronic f-IIP.” 

Introduction

- Page 4, line 16-17; “probably because physicians who do routine check-ups for IPF patients do not usually treat RHs in Western countries”

This is just the authors’ conjecture. The authors should cite some appropriate papers in this manuscript.

Response: Thank you for the valuable comment. To respond the reviewer’s comment, we deleted the following sentences: “The reasons for hospitalization, however, do not appear to have been clarified, probably because physicians who do routine check-ups for IPF patients do not usually treat RHs in Western countries. Patients usually undergo routine check-ups in interstitial lung disease center, however, they admit their local hospitals when acute exacerbation occur.”.

- Page 5, line 5-6; “Identification of the characteristics of IPF patients who require repeated RHs might be useful for prolonging their survival.”

How can the identification of the characteristics contribute to improvement of patients’ survival? The reviewer think that there is a logical gap between the identification and prolonging patients’ survival. More explanation on this should be given in the introduction part.

Response: As the reviewer commented, we think there was a logical gap in the explanation. So, we changed the sentences as follows (P4,L16): “Identification of the characteristics of patients with IPF who require repeated RHs might lead to the development of appropriate prophylactic measures and treatments and a lengthened survival time.” 

Methods

Patients

- The authors did not present inclusion criteria or exclusion criteria in the methods part. Furthermore, in the main body and Tables, there was no description with respect to ILD diagnoses included in this study.

- This study cohort consisted of a variety of ILDs and the treatment regimen was not consistent among the patients or diseases. Therefore, it is difficult to generalize the results of the study.

We clarified the explanation of inclusion and exclusion criteria as follows (P6,L5). We created a new Fig. 1 to summarize it: “The definition of f-IIP was as follows: a chronic form of idiopathic interstitial pneumonia (IIP) that includes IPF and IIP suggestive of IPF but without a definitive diagnosis. The diagnosis of IPF was based on recent guidelines [12]. Patients were excluded if they had other known causes of interstitial lung disease, as follows: domestic or occupational environmental exposure, connective tissue disease, and drug toxicity. Patients with IIP suggestive of IPF but without a definitive diagnosis, who were found to have a pattern of probable usual interstitial pneumonia (UIP) or indeterminate UIP on high-resolution computed tomography (HRCT), were not subjected to surgical lung biopsy. Fig 1 shows the flow chart for exclusion and inclusion.” 

We also added the following sentences to comment on the reviewer’s concern (P22,L4): “Moreover, given that the present study cohort might consist of patients with a variety of fibrotic interstitial lung diseases, and that the treatment regimen was not consistent among the patients or for each type of disease, caution is needed when generalizing the results.”

Diagnosis of acute exacerbation and pulmonary infection

- In this study, the definition of acute exacerbation (AE) of IPF was based on the 2016 international working group report, which was broader than former criteria. This 2016 definition accepts the concept of triggered-AE including infection. Therefore, in the present study, AE may have overlapped with pulmonary infection. I wonder how the authors distinguished between AE and pulmonary infection exhibiting bilateral consolidation and/or ground-glass opacities on HRCT.

Response: Thank you for your important comments. We wrote about the diagnostic criteria in the section “Diagnosis of acute exacerbation and pulmonary infection” and followed these criteria when distinguishing between AE and pulmonary infection. Therefore, obvious cases were easily categorized as AE or pulmonary infection. However, as the reviewer has commented, distinction was difficult in some cases, because some cases overlapped. In those cases, the final diagnosis was made after careful discussions with several specialists. However, we added the following sentences to comment on it as a study limitation (P22,L7). “Finally, distinguishing between acute exacerbation and pulmonary infection was difficult for some cases, even with considerable effort for the differential diagnosis. Acute exacerbation and pulmonary infection might even overlap in some cases.” 

Respiratory-related hospitalization

- Is “heart failure” an acute respiratory event? Please explain why it was included.

Response: There are no standard criteria for an RH. But we included heart failure into RH based on a previous report (Moua T. Chest 2016;149:1205-1214).

Assessment of survival

- Please define in-hospital mortality in this part and show the median (range or IQR) or mean (SD) days.

Response: We added the following sentences to respond to the reviewer’s comment (P8,L18): “In-hospital mortality was defined as percentage of patients who died during the corresponding hospitalization without discharge (mean duration of hospitalization: 33.7±41.1, 26.4±19.4, 22.0±13.1, and 24.3±20.9 days at first, second, third, and ≥fourth hospitalizations, respectively).” 

Results

- Figure 2 was based on the survival time from the date of the first, second, third, or ≥ fourth RH until the date of death or last visit, which is associated with a kind of lead-time bias. Furthermore, patients with multiple hospitalization probably belong to not only the 1st RH group but also ≥2nd RH groups. The 1st, 2nd, 3rd, and ≥4th RH groups were not independent. Therefore, these statistical methods are incorrect.

Response: Patients who underwent 1, 2, 3, and 4 or more hospitalizations during the study period were categorized into “After the first, second, third, or ≥ fourth hospitalization” group. The last hospitalization we recognized in the study period was used for categorization in the Kaplan-Meier curves. So, none of the patients overlapped in the Kaplan-Meier survival curves. We added the following sentence to explain this (P18,L2): “In this analysis, the last hospitalization we recognized during the study period was used for categorization, so that none of the patients overlapped.” 

To avoid misunderstanding, we also added the following at the bottom of the Table 3: “Some patients overlap. For example, a patient who underwent 2 respiratory hospitalizations during the study period was included in the analyses of both the first and second hospitalization.”

We also added the number of patients at risk to Fig. 3.

Discussion

- Page 21, line 9-11; “Preventing hospitalization of patients with f-IIP is crucial for prolonging their survival, especially for patients who have already been hospitalized once for a respiratory cause”.

This sentence sounds strange. Presumably, preventing an acute respiratory event is crucial.

Response: Changed as suggested as follows (P21.L4): “Preventing an acute respiratory event associated with f-IIP is crucial”.

---

## [Decision Letter · Decision Letter 1]

11 Feb 2020

PONE-D-19-28502R1

Characteristics of patients with fibrosing idiopathic interstitial pneumonia undergoing repeated respiratory-related hospitalization: A retrospective cohort study

PLOS ONE

Dear Dr. Nishiyama,

Thank you for submitting your manuscript to PLOS ONE. After careful consideration, we feel that it has merit but does not fully meet PLOS ONE’s publication criteria as it currently stands. Therefore, we invite you to submit a revised version of the manuscript that addresses the points raised during the review process.

For this revised submission I was fortunate to be able to continue with the same two peer reviewers from the initial submission. Whilst both felt there had been some improvements to the manuscript, you will note below some significant ongoing concerns, particularly from Reviewer #2, for your attention.

We would appreciate receiving your revised manuscript by Mar 27 2020 11:59PM. To enhance the reproducibility of your results, we recommend that if applicable you deposit your laboratory protocols in protocols.io, where a protocol can be assigned its own identifier (DOI) such that it can be cited independently in the future. For instructions see: http://journals.plos.org/plosone/s/submission-guidelines#loc-laboratory-protocols

We look forward to receiving your revised manuscript.

Kind regards,

Shane Patman, PhD

Academic Editor

PLOS ONE

Reviewers' comments:

Reviewer's Responses to Questions

**Comments to the Author**

1. If the authors have adequately addressed your comments raised in a previous round of review and you feel that this manuscript is now acceptable for publication, you may indicate that here to bypass the “Comments to the Author” section, enter your conflict of interest statement in the “Confidential to Editor” section, and submit your "Accept" recommendation.

Reviewer #1: All comments have been addressed

Reviewer #2: (No Response)

2. Is the manuscript technically sound, and do the data support the conclusions?

Reviewer #1: Yes

Reviewer #2: Partly

3. Has the statistical analysis been performed appropriately and rigorously? 

Reviewer #1: Yes

Reviewer #2: No

4. Have the authors made all data underlying the findings in their manuscript fully available?

Reviewer #1: Yes

Reviewer #2: Yes

5. Is the manuscript presented in an intelligible fashion and written in standard English?

Reviewer #1: Yes

Reviewer #2: No

6. Review Comments to the Author

Reviewer #1: The authors have respond most of the comments, and only minor corrections are still necessary.

Minor comments

The newly created Figure 1 needs to be revised. Generally, the number of people is required in such a flowchart. For example, the author should list all of the collected patients and their numbers in the first row, and list the excluded patients and their numbers in the second row. I recommend not listing patients in the middle row.

However, the novelty of this paper is that the authors deal with f-IIP patients. This is because previous studies dealing with IPF patients have already been described in the literature. I suggest adding more several explanations of f-IIP in Introduction and Discussion parts.

Reviewer #2: The authors have modified the manuscript and responded to my comments partially. However, for some points, sufficient modification has not been done. Although the study concept is interesting, there are still several critical drawbacks with respect to study design, inclusion and exclusion criteria, heterogeneity of IIPs included in this study, statistical method.

Page 6, line 5-7; “The definition of f-IIP was as follows: a chronic form of idiopathic interstitial pneumonia (IIP) that includes IPF and IIP suggestive of IPF but without a definitive diagnosis.”

1. The authors defined a chronic form of IIP as f-IIP. However, “chronic” f-IIP appeared repeatedly in this manuscript. This is very strange description.

Page 6, line 5-7; “The definition of f-IIP was as follows: a chronic form of idiopathic interstitial pneumonia (IIP) that includes IPF and IIP suggestive of IPF but without a definitive diagnosis.”

Page 6, line 9-12; “Patients with IIP suggestive of IPF but without a definitive diagnosis were patients who were found to have a pattern of probable usual interstitial pneumonia (UIP) or indeterminate UIP on high-resolution computed tomography (HRCT), and were not subjected to surgical lung biopsy.”

2. I am sure that the inclusion of patients with IIP suggestive of IPF but without a definitive diagnosis in this study yields a significant problem because they were just undiagnosed. Such patients definitely include IIPs other than IPF. The proportions of patients with an IIP other than IPF (e.g., NSIP, DIP/RB-ILD, and unclassifiable IIP/ILD) who were included in this study must have affected the results of this study. Furthermore, it is unclear who classified HRCT images from patients into some HRCT patterns (probable UIP, indeterminate for UIP, and others), which also would have affected patient inclusion.

3. I wonder why this study included patients with IIP suggestive of IPF but without definitive diagnosis in addition to those with IPF and why excluded patients with chronic course of IIP who exhibited alternative diagnosis pattern on HRCT, including NSIP, DIP/RB-ILD, PPFE, and unclassifiable IIP/ILD. I think that this inclusion criteria creates a significant bias. If the authors wanted to clarify the clinical significance of RHs in IPF, this study should have included only patients with IPF. If the authors wanted to investigate the real world data of patients with chronic IIPs, this study should have included all patients with chronic IIPs.

Figure 1; Thank you for presenting study flow chart for exclusion and inclusion. However, I might have found some errors.

4. This figure is strange because the exclusion criteria appeared in the upstream of flow chart and the inclusion criteria in the downstream.

Figure 1; “Excluded if the hospitalization is not respiratory related or elective”

5. Is this description right? On the basis of this description, patients who had non-elective hospitalization are excluded from this study because “not respiratory related or elective” has the same meaning as “neither respiratory related nor elective”.

Page 9, line 3-4; “The total survival time from the first day of respiratory-related hospitalization”

6. This description is confusing. Does this mean survival time from the first day of the first respiratory-related hospitalization until last visit or death, or the survival time from the first day of “each” respiratory-related hospitalization until last visit or death (e.g., the survival time from the first day of the second respiratory-related hospitalization until last visit or death) ? This is a serious concern, which is related to my previous reviewer comment, because this definition is significantly associated with Figure 3.

Page 10, line 3 and Figure 2;

7. I was confused with this Figure. What do these transverse lines mean? Sufficient details are not given in this manuscript or figure legend.

8. “Hospital death” and “died of non-respiratory causes” appeared in this figure. Did these overlap?

Page 18, line 2-5 and Figure 3; “In this analysis, the last hospitalization we recognized during the study period was used for categorization, so that none of the patients overlapped. Survival after the first hospitalization was significantly better than the survival of patients after multiple hospitalizations.”

9. This is related to the previous review comment and my current question No 6. On the basis of the survival time from the first day of each RH (the first, second, third, or ≥fourth) until the date of death or last visit, this figure represents a kind of lead-time bias, which is misleading for readers.

10. The authors explained that none of the patients that were categorized into 4 groups (after 1st, 2nd, 3rd, and 4th hospitalization) overlapped. Does this mean that patients in the after 1st hospitalization group underwent RH only once during the study period? Did patients in the after 2nd, those in the after 3rd, and those in the after ≥ 4th hospitalization groups underwent RH just twice, just three times, and just four times, respectively, during the study period? If so, I wonder why there is the discordance between the numbers of patient shown in Figure 2 and those in Figure 3. For example, with respect to the after 1st hospitalization group, the number of patients in Figure 3 is 120. However, in Figure 2, the number of patient in 1st RH (n = 243) minus that in 2nd RH (n = 127) is 116.

Page 19, line 16-page 20, line 2; “However, there exists some doubt about the reasons for hospitalization and death in patients with IPF, because physicians who perform routine check-ups of IPF patients do not usually treat RHs in Western countries. Patients usually undergo routine check-ups at an interstitial lung disease center; however, each patient is admitted to his or her local hospital when an acute exacerbation occurs.”

11. This is just the authors’ conjecture. The authors should cite some appropriate papers in this manuscript. In the previous review, I made a similar comment.

Page 21, line 4-5; “Preventing an acute respiratory event associated with f-IIP is crucial for prolonging survival”

12. This statement is not supported by any data/demonstration in this study.

7. PLOS authors have the option to publish the peer review history of their article (what does this mean?). If published, this will include your full peer review and any attached files.

Reviewer #1: No

Reviewer #2: No

---

## [Author Response · Author response to Decision Letter 1]

21 Feb 2020

Dear Dr. Patman:

Thank you for your kind and careful review of our submitted revised manuscript. We tried to address all of the reviewers' concerns. In addition, we revised the title somewhat.

To Reviewer #1: 

Minor comments

The newly created Figure 1 needs to be revised. Generally, the number of people is required in such a flowchart. For example, the author should list all of the collected patients and their numbers in the first row, and list the excluded patients and their numbers in the second row. I recommend not listing patients in the middle row.

Response: Revised as suggested.

However, the novelty of this paper is that the authors deal with f-IIP patients. This is because previous studies dealing with IPF patients have already been described in the literature. I suggest adding more several explanations of f-IIP in Introduction and Discussion parts.

Response: In accordance with the comments of another reviewer, we changed the “f-IIP” to “chronic IIP (c-IIP)” including in the title. However, to respond to the comments of the Reviewer #1, we added the following text to the Introduction (P5,L1) and the Discussion (P22,L4): “Patients with a chronic form of idiopathic interstitial pneumonia (IIP) other than IPF are common. They have characteristics suggestive of IPF but are without a definitive diagnosis because surgical lung biopsy was not performed for some reason. They should be classified as patients with unclassifiable IIP [12]. However, in general practice, it may be relevant to consider patients with IPF together with patients with unclassifiable fibrotic IIP” and “Actually, the patients with c-IIPs other than IPF who were included in the study corresponded to those with unclassifiable IIP, which was defined by an official 2013 American Thoracic Society/European Respiratory Society statement [12]. The concept of unclassifiable IIP needs to be refined in the future.”

Reviewer #2: The authors have modified the manuscript and responded to my comments partially. However, for some points, sufficient modification has not been done. Although the study concept is interesting, there are still several critical drawbacks with respect to study design, inclusion and exclusion criteria, heterogeneity of IIPs included in this study, statistical method.

Page 6, line 5-7; “The definition of f-IIP was as follows: a chronic form of idiopathic interstitial pneumonia (IIP) that includes IPF and IIP suggestive of IPF but without a definitive diagnosis.”

1. The authors defined a chronic form of IIP as f-IIP. However, “chronic” f-IIP appeared repeatedly in this manuscript. This is very strange description.

Response: Thank you for your valuable comments. We changed the term “fibrotic IIP (f-IIP)” to “chronic IIP (c-IIP) including in the title. Patients with c-IIP included those with an IPF diagnosis and those with probable UIP or an indeterminate pattern on chest HRCT. The latter 2 corresponds to unclassifiable IIP, as defined by the 22013 ATS/ERS IIP, because there were no patients who were pathologically diagnosed in the latter group of patients. 

Page 6, line 5-7; “The definition of f-IIP was as follows: a chronic form of idiopathic interstitial pneumonia (IIP) that includes IPF and IIP suggestive of IPF but without a definitive diagnosis.”

Page 6, line 9-12; “Patients with IIP suggestive of IPF but without a definitive diagnosis were patients who were found to have a pattern of probable usual interstitial pneumonia (UIP) or indeterminate UIP on high-resolution computed tomography (HRCT), and were not subjected to surgical lung biopsy.”

2. I am sure that the inclusion of patients with IIP suggestive of IPF but without a definitive diagnosis in this study yields a significant problem because they were just undiagnosed. Such patients definitely include IIPs other than IPF. The proportions of patients with an IIP other than IPF (e.g., NSIP, DIP/RB-ILD, and unclassifiable IIP/ILD) who were included in this study must have affected the results of this study. Furthermore, it is unclear who classified HRCT images from patients into some HRCT patterns (probable UIP, indeterminate for UIP, and others), which also would have affected patient inclusion.

Response: We changed the term “f-IIP” to “c-IIP”. At the same time, the definition of c-IIP other than IPF was also changed as follows (P6,L8); “Patients with c-IIP other than IPF were patients who were found to have a pattern of probable usual interstitial pneumonia (UIP) or indeterminate UIP on high-resolution computed tomography (HRCT), and were not subjected to surgical lung biopsy. Patients with an alternative diagnosis pattern on chest HRCT were excluded. HRCT patterns were categorized by pulmonologists with over 10 years of experience. For patients whose HRCT pattern was difficult to categorize, the final assessment was established after careful discussions between several specialists.”. Because we wanted to focus on IPF and IIP suggestive of IPF, patients with an alternative diagnostic pattern on chest HRCT were excluded from the study. However, we deleted the expression of “IPF and IIP suggestive of IPF” from the text in our manuscript to avoid confusion.

3. I wonder why this study included patients with IIP suggestive of IPF but without definitive diagnosis in addition to those with IPF and why excluded patients with chronic course of IIP who exhibited alternative diagnosis pattern on HRCT, including NSIP, DIP/RB-ILD, PPFE, and unclassifiable IIP/ILD. I think that this inclusion criteria creates a significant bias. If the authors wanted to clarify the clinical significance of RHs in IPF, this study should have included only patients with IPF. If the authors wanted to investigate the real world data of patients with chronic IIPs, this study should have included all patients with chronic IIPs.

Response: Thank you for your valuable suggestion. As we responded to comment 2, the term “f-IIP” was changed to “c-IIP” to include all patients with chronic IIPs. 

Figure 1; Thank you for presenting study flow chart for exclusion and inclusion. However, I might have found some errors.

4. This figure is strange because the exclusion criteria appeared in the upstream of flow chart and the inclusion criteria in the downstream.

Response: Changed as suggested. We also changed some expressions in Fig. 1 according to a native speaker’s recommendation.

Figure 1; “Excluded if the hospitalization is not respiratory related or elective”

5. Is this description right? On the basis of this description, patients who had non-elective hospitalization are excluded from this study because “not respiratory related or elective” has the same meaning as “neither respiratory related nor elective”.

Response: Changed as follows: “Excluded for nonrespiratory-related and/or elective hospitalizations”.

Page 9, line 3-4; “The total survival time from the first day of respiratory-related hospitalization”

6. This description is confusing. Does this mean survival time from the first day of the first respiratory-related hospitalization until last visit or death, or the survival time from the first day of “each” respiratory-related hospitalization until last visit or death (e.g., the survival time from the first day of the second respiratory-related hospitalization until last visit or death) ? This is a serious concern, which is related to my previous reviewer comment, because this definition is significantly associated with Figure 3.

Response: Thank you for your suggestion regarding this important definition. Survival time from each respiratory-related hospitalization is correct. So, we changed it as follows (P9,L6): “The total survival time from the first day of each respiratory-related hospitalization was also determined.”.

Page 10, line 3 and Figure 2;

7. I was confused with this Figure. What do these transverse lines mean? Sufficient details are not given in this manuscript or figure legend.

Response: We revised Fig 2. In addition, we added the following explanation (P10,L3): “As shown in Fig 2, of 243 patients who were hospitalized for the first time, 39 (16.0%) died in the hospital. Of 204 patients (84.0%) who were discharged, 15 (6.2%) were not followed-up, 50 (20.6%) were followed-up without any RH, 12 (4.9%) died of nonrespiratory-related causes, and 127 (52.3) underwent a second RH during the period of observation. Fig 2 also shows the distribution of patients after the second RH.”

8. “Hospital death” and “died of non-respiratory causes” appeared in this figure. Did these overlap?

Response: No, they did not overlap. “Death due to nonrespiratory cause” indicates patients who died of nonrespiratory-related causes after discharge. We revised Fig 2 and added a clarified explanation (P10,L3).

Page 18, line 2-5 and Figure 3; “In this analysis, the last hospitalization we recognized during the study period was used for categorization, so that none of the patients overlapped. Survival after the first hospitalization was significantly better than the survival of patients after multiple hospitalizations.”

9. This is related to the previous review comment and my current question No 6. On the basis of the survival time from the first day of each RH (the first, second, third, or ≥fourth) until the date of death or last visit, this figure represents a kind of lead-time bias, which is misleading for readers.

Response: To avoid misleading the readers, we added following (P20,L17): “although lead-time bias should be taken into account”.

10. The authors explained that none of the patients that were categorized into 4 groups (after 1st, 2nd, 3rd, and 4th hospitalization) overlapped. Does this mean that patients in the after 1st hospitalization group underwent RH only once during the study period? Did patients in the after 2nd, those in the after 3rd, and those in the after ≥ 4th hospitalization groups underwent RH just twice, just three times, and just four times, respectively, during the study period? If so, I wonder why there is the discordance between the numbers of patient shown in Figure 2 and those in Figure 3. For example, with respect to the after 1st hospitalization group, the number of patients in Figure 3 is 120. However, in Figure 2, the number of patient in 1st RH (n = 243) minus that in 2nd RH (n = 127) is 116.

Response: Thank you for your comments. We revised the numbers in Fig 3.

Page 19, line 16-page 20, line 2; “However, there exists some doubt about the reasons for hospitalization and death in patients with IPF, because physicians who perform routine check-ups of IPF patients do not usually treat RHs in Western countries. Patients usually undergo routine check-ups at an interstitial lung disease center; however, each patient is admitted to his or her local hospital when an acute exacerbation occurs.”

11. This is just the authors’ conjecture. The authors should cite some appropriate papers in this manuscript. In the previous review, I made a similar comment.

Response: We agree with the reviewer’s opinion. So, we deleted the relevant text.

Page 21, line 4-5; “Preventing an acute respiratory event associated with f-IIP is crucial for prolonging survival”

12. This statement is not supported by any data/demonstration in this study.

Response: We revised this expression as follows (P21,L1): “Preventing an acute respiratory event associated with c-IIP may be another challenge for the prolongation of survival,”.

---

## [Decision Letter · Decision Letter 2]

17 Mar 2020

PONE-D-19-28502R2

Characteristics of patients with chronic idiopathic interstitial pneumonia undergoing repeated respiratory-related hospitalization: A retrospective cohort study

PLOS ONE

Dear Dr. Nishiyama,

Thank you for submitting your manuscript to PLOS ONE. After careful consideration, we feel that it has merit but does not fully meet PLOS ONE’s publication criteria as it currently stands. Therefore, we invite you to submit a revised version of the manuscript that addresses the points raised during the review process.

I have been fortunate in being able to continue with the same two expert peer reviewers for this latest cycle of review. Again positive changes to the manuscript have been noted, but the reviewers have a few minor recommendations for addressing, as outlined below.

We would appreciate receiving your revised manuscript by May 01 2020 11:59PM. To enhance the reproducibility of your results, we recommend that if applicable you deposit your laboratory protocols in protocols.io, where a protocol can be assigned its own identifier (DOI) such that it can be cited independently in the future. For instructions see: http://journals.plos.org/plosone/s/submission-guidelines#loc-laboratory-protocols

We look forward to receiving your revised manuscript.

Kind regards,

Shane Patman, PhD

Academic Editor

PLOS ONE

Reviewers' comments:

Reviewer's Responses to Questions

**Comments to the Author**

1. If the authors have adequately addressed your comments raised in a previous round of review and you feel that this manuscript is now acceptable for publication, you may indicate that here to bypass the “Comments to the Author” section, enter your conflict of interest statement in the “Confidential to Editor” section, and submit your "Accept" recommendation.

Reviewer #1: (No Response)

Reviewer #2: (No Response)

2. Is the manuscript technically sound, and do the data support the conclusions?

Reviewer #1: Partly

Reviewer #2: Partly

3. Has the statistical analysis been performed appropriately and rigorously? 

Reviewer #1: I Don't Know

Reviewer #2: I Don't Know

4. Have the authors made all data underlying the findings in their manuscript fully available?

Reviewer #1: No

Reviewer #2: Yes

5. Is the manuscript presented in an intelligible fashion and written in standard English?

Reviewer #1: Yes

Reviewer #2: No

6. Review Comments to the Author

Reviewer #1: The author only changed "f-IIP" to "c-IIP" in this revision. It is unclear why the authors have not used more common term such as "fibrosing interstitial lung disease (f-ILD)" with numerous previous literatures. Furthermore, the term" unclassifiable fibrotic IIP" seems to be unusual. The author would need to provide the details of them and several references to support the validity of these new terms.

Reviewer #2: I appreciate the authors’ responses. This manuscript has been improved. I think this manuscript will be acceptable after some corrections have been done.

1. The inclusion and exclusion criteria (Figure 1) is probably incorrect. I think that, in this study, the inclusion criteria is patients who were classified/diagnosed with a chronic form of ILD including IPF and were hospitalized for the first time during the study period (n=1028), and patients were excluded from this study if any of the following criteria were met: exhibiting alternative diagnosis pattern on HRCT (n=127); a diagnosis with a secondary cause of ILD (n=294); and nonrespiratory-related and/or elective hospitalizations (n=364). Consequently, 243 patients with c-IIP were enrolled in this study (138 patients with IPF and 105 with a chronic form of ILD who exhibited probable UIP or indeterminate for UIP on HRCT). Please amend.

2. Diagnosis of acute exacerbation: based on this definition, only patients with a chronic form of ILD who exhibited UIP pattern on HRCT can be diagnosed with acute exacerbation. However, this study included not only such patients but also those who exhibited probable/indeterminate for UIP on HRCT. Please amend this definition.

3. Respiratory-related hospitarization: “RH was defined as a nonelective hospitalization due to an acute respiratory event that included acute exacerbation of IPF.” “IPF” should be replaced by “c-IIP”.

4. I do suggest to have some external language editing done by a person familiar with this field.

7. PLOS authors have the option to publish the peer review history of their article (what does this mean?). If published, this will include your full peer review and any attached files.

Reviewer #1: No

Reviewer #2: No

---

## [Author Response · Author response to Decision Letter 2]

30 Mar 2020

Dear Dr. Patman:

Thank you for your kind and careful review of our submitted revised manuscript. We again tried to address all of the reviewers' concerns.

To Reviewer #1: 

The author only changed "f-IIP" to "c-IIP" in this revision. It is unclear why the authors have not used more common term such as "fibrosing interstitial lung disease (f-ILD)" with numerous previous literatures. Furthermore, the term" unclassifiable fibrotic IIP" seems to be unusual. The author would need to provide the details of them and several references to support the validity of these new terms.

Response: 

We changed “f-IIP” to “c-IIP” according to a suggestion of Reviewer 2. “Fibrosing ILD” includes ILD other than IIP, such as chronic hypersensitive pneumonitis and collagen vascular disease related ILD. Actually, we used the term c-IIP, because we wanted to include patients with IPF and IIP suggestive of IPF but without a definitive diagnosis. But, as Reviewer 2 suggested, the term c-IIP has not been validated. Therefore, we added the following sentences in the limitation part of the Discussion section citing several reports using similar terms (P23, L3): “We defined a patient with c-IIP as a patient with a diagnosis of IPF or with an HRCT pattern of probable or indeterminate for UIP. The term “c-IIP” has not been validated, although it has been used in several studies [22-23].”. And we changed a sentence as follows (P23, L8): “The concepts of c-IIP and unclassifiable IIP need further refinement.”

To Reviewer #2: 

I appreciate the authors’ responses. This manuscript has been improved. I think this manuscript will be acceptable after some corrections have been done.

1. The inclusion and exclusion criteria (Figure 1) is probably incorrect. I think that, in this study, the inclusion criteria is patients who were classified/diagnosed with a chronic form of ILD including IPF and were hospitalized for the first time during the study period (n=1028), and patients were excluded from this study if any of the following criteria were met: exhibiting alternative diagnosis pattern on HRCT (n=127); a diagnosis with a secondary cause of ILD (n=294); and nonrespiratory-related and/or elective hospitalizations (n=364). Consequently, 243 patients with c-IIP were enrolled in this study (138 patients with IPF and 105 with a chronic form of ILD who exhibited probable UIP or indeterminate for UIP on HRCT). Please amend.

Response: Amended as suggested.

2. Diagnosis of acute exacerbation: based on this definition, only patients with a chronic form of ILD who exhibited UIP pattern on HRCT can be diagnosed with acute exacerbation. However, this study included not only such patients but also those who exhibited probable/indeterminate for UIP on HRCT. Please amend this definition.

Response: We actually explained this in the previous manuscript as follows: “The diagnosis of acute exacerbation of c-IIP was made in reference to these criteria.”. However, we amended it to be easily understood as follows (P7, L6): “Acute exacerbation of c-IIP was defined in reference to a recent international working group report for acute exacerbation of IPF [14] as follows: 1) a previous or concurrent diagnosis of c-IIP; 2) acute worsening or development of dyspnea typically of less than 1 month’s duration; 3) HRCT finding of new bilateral ground-glass opacities and/or consolidation superimposed on a background consistent with the UIP, probable UIP, or indeterminate for UIP pattern; and 4) deterioration not fully explained by cardiac failure or fluid overload.”.

3. Respiratory-related hospitarization: “RH was defined as a nonelective hospitalization due to an acute respiratory event that included acute exacerbation of IPF.” “IPF” should be replaced by “c-IIP”.

Response: Revised as suggested.

4. I do suggest to have some external language editing done by a person familiar with this field.

Response: We requested external language editing again. Therefore, some grammatical revisions were made.

---

## [Decision Letter · Decision Letter 3]

10 Apr 2020

Characteristics of patients with chronic idiopathic interstitial pneumonia undergoing repeated respiratory-related hospitalization: A retrospective cohort study

PONE-D-19-28502R3

Dear Dr. Nishiyama,

We are pleased to inform you that your manuscript has been judged scientifically suitable for publication and will be formally accepted for publication once it complies with all outstanding technical requirements.

With kind regards,

Shane Patman, PhD

Academic Editor

PLOS ONE

Additional Editor Comments (optional):

Reviewers' comments:

Reviewer's Responses to Questions

**Comments to the Author**

1. If the authors have adequately addressed your comments raised in a previous round of review and you feel that this manuscript is now acceptable for publication, you may indicate that here to bypass the “Comments to the Author” section, enter your conflict of interest statement in the “Confidential to Editor” section, and submit your "Accept" recommendation.

Reviewer #1: All comments have been addressed

Reviewer #2: All comments have been addressed

2. Is the manuscript technically sound, and do the data support the conclusions?

Reviewer #1: Yes

Reviewer #2: Yes

3. Has the statistical analysis been performed appropriately and rigorously? 

Reviewer #1: I Don't Know

Reviewer #2: I Don't Know

4. Have the authors made all data underlying the findings in their manuscript fully available?

Reviewer #1: Yes

Reviewer #2: Yes

5. Is the manuscript presented in an intelligible fashion and written in standard English?

Reviewer #1: Yes

Reviewer #2: Yes

6. Review Comments to the Author

Reviewer #1: I think that the authors have properly respond to the comments from the reviewers, and have modified their manuscript.

Reviewer #2: (No Response)

7. PLOS authors have the option to publish the peer review history of their article (what does this mean?). If published, this will include your full peer review and any attached files.

Reviewer #1: No

Reviewer #2: No

---

## [Editor Report · Acceptance letter]

14 Apr 2020

PONE-D-19-28502R3 

Characteristics of patients with chronic idiopathic interstitial pneumonia undergoing repeated respiratory-related hospitalizations: A retrospective cohort study 

Dear Dr. Nishiyama:

I am pleased to inform you that your manuscript has been deemed suitable for publication in PLOS ONE. Congratulations! Your manuscript is now with our production department. 

With kind regards,

on behalf of

Assoc Prof Shane Patman 

Academic Editor

PLOS ONE